# Sub-Perception and Supra-Perception Spinal Cord Stimulation in Chronic Pain Syndrome: A Randomized, Semi-Double-Blind, Crossover, Placebo-Controlled Trial

**DOI:** 10.3390/jcm9092810

**Published:** 2020-08-31

**Authors:** Paweł Sokal, Agnieszka Malukiewicz, Sara Kierońska, Joanna Murawska, Cezary Guzowski, Marcin Rudaś, Dariusz Paczkowski, Marcin Rusinek, Mateusz Krakowiak

**Affiliations:** 1Department of Neurosurgery and Neurology, Jan Biziel University Hospital Nr 2, Ujejskiego 75 Street, 85-168 Bydgoszcz, Poland; agnieszka.malukiewicz@gmail.com (A.M.); sara.kieronska@biziel.pl (S.K.); marcin.rudas@biziel.pl (M.R.); dariusz.paczkowski@biziel.pl (D.P.); marcin.rusinek@biziel.pl (M.R.); mateusz.krakowiak@biziel.pl (M.K.); 2Faculty of Health Sciences, Collegium Medicum in Bydgoszcz Nicolaus Copernicus University in Toruń, Jagielońska 13-15 85-067 Bydgoszcz, Poland; 3Students’ Scientific Circle at the Department of Neurosurgery, Jan Biziel University Hospital Nr 2, Ujejskiego 75 Street, 85-168 Bydgoszcz, Poland; joanna.murawska94@wp.pl (J.M.); cezary.guzowski@gmail.com (C.G.)

**Keywords:** spinal cord stimulation, failed back surgery syndrome, complex regional pain syndrome, cross-over trial, placebo control

## Abstract

Background: The introduction of modern sub-perception modalities has improved the efficacy of spinal cord stimulation (SCS) in refractory pain syndromes of the trunk and lower limbs. The objective of this study was to evaluate the effectiveness of low and high frequency SCS among patients with chronic pain. Material and methods: A randomised, semi-double-blind, placebo controlled, four period (4 × 2 weeks) crossover trial was conducted from August 2018 to January 2020. Eighteen patients with SCS due to failed back surgery syndrome and/or complex regional pain syndrome were randomised to four treatment arms without washout periods: (1) low frequency (40-60 Hz), (2) 1 kHz, (3) clustered tonic, and (4) sham SCS (i.e., placebo). The primary outcome was pain scores measured by visual analogue scale (VAS) preoperatively and during subsequent treatment arms. Results: Pain scores (VAS) reported during the preoperative period was M (SD) = 8.13 (0.99). There was a 50% reduction in pain reported in the low frequency tonic treatment group (M (SD) = 4.18 (1.76)), a 37% reduction in the 1 kHz treatment group (M (SD) = 5.17 (1.4)), a 34% reduction in the clustered tonic settings group (M (SD) = 5.27 (1.33)), and a 34% reduction in the sham stimulation group (M (SD) = 5.42 (1.22)). The reduction in pain from the preoperative period to the treatment period was significant in each treatment group (*p* < 0.001). Overall, these reductions were of comparable magnitude between treatments. However, the modality most preferred by patients was low frequency (55% or 10 patients). Conclusions: The pain-relieving effects of SCS reached significance and were comparable across all modes of stimulation including sham. Sub-perception stimulation was not superior to supra-perception. SCS was characterised by a high degree of placebo effect. No evidence of carryover effect was observed between subsequent treatments. Contemporary neuromodulation procedures should be tailored to the individual preferences of patients.

## 1. Introduction

Patients with chronic pain who are not effectively treated with standard conservative medical therapy are candidates for neuromodulation therapy. Spinal cord stimulation (SCS) is an important neuromodulation tool for the management of chronic pain conditions, such as failed back surgery syndrome (FBSS), complex regional pain syndrome (CRPS), peripheral diabetic neuropathy (PDN), and other types of chronic low back and/or lower limb pain syndromes [1,2,3,4]. The recognised benefits of SCS include reductions in pain, anxiety, and depression, which may lead to improved quality of life (QoL) but not necessarily a reduction of occupational disability [1,5,6,7,8]. Conventional systems applied an electrical field on the spinal cord with a frequency of less than 100 Hz. A meta-analysis showed that the effectiveness of these systems was over 50% in 58% of patients [3,9]. Recent advancements in the technology of SCS have allowed for the delivery of stimulation that is free of paraesthesia. For example, De Ridder et al. introduced burst stimulation, which is generally paraesthesia free and characterised by high-frequency bursts (500 Hz) followed by low-frequency intervals [10]. The largest randomised controlled trial of SCS, the SUNBURST study, demonstrated that burst SCS was more effective in reducing pain than tonic SCS [11]. High frequency (HF) stimulation with a frequency of 10 kHz showed a sustained 24-month superiority over traditional SCS [12]. SENZA-RTC confirmed that high-frequency SCS that produced no sensation was superior to low-frequency stimulation for the treatment of chronic pain of the trunk and limbs. Sub-perception neuromodulation over 1 kHz has proven to be effective for relieving pain across several studies [12,13].

The information about comparative efficiency of different modes of stimulation is scanty despite several published reports in this area [14,15,16,17]. Sensation-free modes of SCS provide a unique opportunity to conduct a placebo-controlled trial in a crossover design and to assess the power of sham stimulation [18].

The objective of this study was to evaluate the effectiveness of currently available types of SCS. We tested three available waveforms: (1) low frequency (LF), (2) 1 kHz, and (3) clustered tonic. We additionally tested sham stimulation to assess the placebo effect. We aimed to examine the effects of various stimulation modes on levels of pain and disability, as well as, assess the amount of analgesic medication required during each mode.

## 2. Experimental Section

### 2.1. Materials and Methods

This study was conducted according to the rules of the Helsinki Declaration and requirements of the local ethics committee. Permission nr 629/2018 has been given by Bioethics Committee of the Nicolaus Copernicus University in Bydgoszcz. Each patient was informed about the details of the study and provided written informed consent. The trial was registered in the appropriate registry (ClinicalTrials.gov Identifier: NCT03957395).

Key inclusion criteria consisted of (1) patients with FBSS or CRPS with neuropathic and mixed pain in the low-back and/or legs that is refractory to conservative therapy, (2) chronic pain reported for at least 6 months, and (3) 18–80 years of age. Exclusion criteria were (1) active malignancy, (2) addiction to alcohol and/or medication, (3) evidence of an active disruptive psychiatric disorder, (4) local infection at the site of surgical incision, and (5) pregnancy.

Twenty-three subjects (11 women) with chronic pain were enrolled in the study. Patients ranged in age from 35 to 74 years (see Table 1). Eighteen patients (78%) were diagnosed with FBSS and five (22%) were diagnosed with CRPS, and pain was distributed in the lumbosacral area. Patients were recruited in the Department of Neurosurgery and Neurology of University Hospital nr 2 of Collegium Medicum of Nicolaus Copernicus University in Bydgoszcz, Poland. Mean duration of chronic pain was 8.3 years (range 1–30 years).

### 2.2. Implantations

Examined patients underwent percutaneous implantation of either one or two linear lead 8- or 16-contact (Infinion 16^TM^) electrodes on levels Th7–Th10 for low back and limbs pain. Surgeries were conducted in two stages. The first stage was performed using local anaesthesia with intraoperative testing followed by 2 weeks of trial stimulation. During the trial period, tonic LF stimulation was used to check the coverage of pain area with paraesthesia induced by an external stimulator by adjusting the optimal settings of active electrode’s contacts. After a successful 14-day trial period, participants who achieved at least a 50% reduction in pain were qualified to the second stage of the study, which involved the placement of a permanent internal pulse generator (IPG) implantation under general anaesthesia. Patients received a non-rechargeable IPG (Precision Novi^TM^) and in one case (patient 11), a rechargeable IPG (Montage^TM^) that was produced by Boston Scientific Co., Boston, MA, USA. The other group consisted of patients who could not undergo percutaneous implantation. These patients were qualified to receive a one-stage surgery that included the implantation of a surgical paddle electrode (Artisan^TM^) with central flavectomy on the thoracic level and simultaneous implantation of a permanent IPG in the subcutaneous pocket. 

This study reported on a total of 23 participants. Five participants were subsequently excluded from analyses for the following reasons: one patient failed a trial period after percutaneous electrode implantation (i.e., did not achieve satisfactory pain relief; patient 21); three participants (patients 1, 22, and 23) did not agree to further evaluation; and in one participant, feedback data were deemed irrelevant and unreliable (patient 12).

Thus, 18 participants entered the randomisation phase two weeks after last surgery. All 18 participants were evaluated for a minimum of eight weeks in a prospective randomised trial. Participants were asked to rate their pain intensity using a visual analogue scale (VAS) upon trial entry and as a primary outcome measure. Participants also completed the Oswestry Disability Index (ODI) as a secondary outcome measure. Patient 11, who had the rechargeable IPG, was asked to charge it once per week, for half an hour each charge, despite the battery status. Ultimately, a total of 13 subjects who underwent assessment during the trial and 5 additional participants who received permanent implants without the trial period were randomised to one of four treatment arms: (1) LF tonic, (2) 1 kHz, (3) clustered tonic, or (4) sham stimulation.

### 2.3. Programming and Randomisation

The study was conducted according to the rules of observational, randomised, semi-double-blind, crossover, controlled study in line with recommendations presented in CONSORT 2010 statement [19]. All subjects were blindly randomised to treatment modality. The trial had 4 arms of treatment that was allocated randomly: (1) LF, (2) 1 kHz, (3) clustered tonic, and (4) sham stimulation. Randomisation was performed by drawing notes with the name of the modality or treatment arm (i.e., LF tonic, 1kHz, clustered tonic, sham) by an independent examiner. At the next meeting (after two weeks), the selection of notes with names of modalities was modified by the notes that were previously chosen. Each blinded phase lasted two weeks. Patients assigned to the LF treatment arm received tonic stimulation with frequencies typically between 40–60 Hz. The pulse width (PW) in the LF treatment arm ranged between 250–500 μs, and the amplitude produced comfortable paraesthesia. In burst stimulation, the same patients received intermittent packets of burst stimuli delivered using the neural targeting algorithm, which consisted of several pulses per packet with PW 250–500 μs repeated with f = 40 Hz. Target amplitude was tailored to patient comfort level and at 50% below perception in a continuous mode. The 1kHz waveform was programmed with f = 1 kHz, PW = 120 μs, and amplitude = 3 Amp. Stimulation of 1kHz was below perceptual threshold (i.e., ± 6 Amp). Crossover took a total of 8 weeks and consisted of four 2-week periods of (1) LF, (2) clustered tonic, (3) 1 kHz, and (4) sham (inactive) stimulation, each. During the control arm (i.e., sham), IPG was deactivated except for emergency shutdown of stimulation. IPG was set on the settings f = 2 Hz, PW = 10 μs, and amp = 0.1 mA. Patients had their own handheld programmers, and the amplitude on the patient’s programmer was adjusted up to 100% so that the patient could clearly see that the IPG was on. During each 2-week stimulation period, individuals reported their pain intensity levels three times each day in a diary. They also recorded the number and type of medications taken. At the end of the randomisation phase, the mode of stimulation that was most effective was revealed to the patients. The independent representative was responsible for the programming and allocation of the stimulation paradigm. The consulting physician present during each visit did not know the kind of waveform that had been programmed by the representative. Thus, participants in the study and the examining physician were both blinded to the type of waveform applied. Stimulators were programmed in such a way that patients did not feel paraesthesia in three of the four modes (1 kHz, clustered tonic, and sham). Participants were programmed in a sitting position without checking the coverage area, given that the coverage check was already conducted in an external trial stimulation 2 weeks prior to randomisation. At the end of the study, the researcher analysing the data and the statistician received reports regarding the programmed waveforms only at select periods of time. The CONSORT flowchart (Figure 1) shows the enrolment of subjects, distinct phases of the trial, and flow of patients.

### 2.4. Data Analyses

Statistical analyses and data manipulation and visualisation was conducted using R 3.6.2 statistical environment [20]. To determine the effects of treatment type, we used Bayesian multilevel regression models implemented in the brms package [21,22]. We also used linear models for average daily pain and a zero-inflated negative binomial for total number of drugs taken. For both models, we used a stepwise procedure of testing the effects. First, a null model was fit to the data with the entire set of subject-level effects and population-level intercept. Second, main effects of the tested factors were added to the model (the “ME model”). Finally, interaction terms were added, resulting in the full model. Goodness of fit of each model was measured with leave-one-out information criterion (LOOIC) [23]. Lower values of LOOIC indicate better model fit, and a significant difference between two models occurs when the absolute difference in LOOICs of the models exceeds two times the standard error of the difference [24].

In Bayesian statistics, the goal is to estimate posterior probability distribution of model parameters by integrating likelihood with prior probability distribution of the values of the parameters. We used default priors implemented in brms, which are weakly informative and do not exert much influence onto parameter estimates [22]. Inference about parameters is conducted by summarising the posterior distribution as a mean and 95% confidence intervals (i.e., 95% CI). The effect captured by a parameter is considered to be statistically significant when the corresponding 95% CI exclude zero [25]. To approximate the posterior distribution, the brms package uses NUTS sampler [26] written in STAN language [27] (see Appendix A for details).

To test the change in average pain from baseline levels, one sample *t* tests were used on difference scores (i.e., average VAS–baseline VAS scores). Bonferroni correction was applied to *p* values to control for multiple comparisons. To test changes in self-reported pain from baseline, we first computed average VAS scores separately for each treatment and subject. Pain levels were averaged across treatment day given that no effects of treatment day were observed. We then subtracted these individual means from individual VAS ratings measured during baseline—hereafter ΔVAS. Due to missing data, for some treatment types between participants, mean ΔVAS scores for each treatment were tested against zero using separate *t* tests.

Change in Oswestry scores were tested using the nonparametric Wilcoxon test.

## 3. Results

### 3.1. Effects of Treatment on Daily Pain Levels

To investigate changes in daily pain during the four tested treatments, we analysed average daily pain for each. Average daily pain was computed by averaging the ratings provided in the mornings, afternoons, and evening, separately for each day, treatment, and subject. Using descriptive analyses, no daily trends were evident in the data on the individual subject level (Figure 2). We did observe substantial between-subject variability in average pain levels within and between treatments. We also observed small variability between daily trends. 

We found that neither the ME model nor full model had a better fit to the data than the null model, ΔLOOIC_ME-Null_ = 0.55 (2.34), and ΔLOOIC_Full-Null_ = 8.04 (3.31), respectively. These results indicate that average pain did not differ significantly between treatments and was relatively stable throughout the entire trial. Parameter estimates for the ME model are presented in Table A1, and marginal predicted means for average pain are presented for each treatment type separately, in Figure 3.

### 3.2. Change in Pain Levels from Baseline

We observed a significant reduction in self-reported pain for each treatment type, and the magnitude of the observed effects were large (Table 2).

Figure 4 presents the distribution of ΔVAS scores separately for each treatment type. Overall, the treatment-related reductions in pain were of comparable magnitude between treatments, including placebo.

The LF mode appeared to result in a slightly greater degree of pain reduction, but this may be driven by two participants who reported unusually low levels of pain (see Figure 2; participants 2 and 3).

After the four-period randomisation, LF stimulation was most preferred by 10 patients (55%); 1 kHz stimulation was most effective for 38% (*n* = 7); and clustered tonic mode was most effective for 5% (*n* = 1) of patients.

### 3.3. Effects of Treatments on the Number and Type of Medications

To test for differences in drug intake between treatments, we calculated the total number of drugs taken during each treatment period by each subject (Figure 5).

The ME1 model consisted of the main effect of drug type; the ME2 model consisted of the main effects of drug and treatment types; and the full model consisted of main effects and interactions between drug and treatment types. We found that the ME1 model had the best fit to the data. The ME1 model was also the only model that had better fit than the null model, ΔLOOIC_ME1-Null_ = −8.8 (7.6). These results indicate that the treatments did not significantly differ from each other in terms of total number of medications taken. However, we observed credible differences in total number of pills taken between medications (see Table A2 for parameter estimates of the ME1 model). The anticonvulsants were most frequently not taken at all (*z* = 81%, (71%, 88%)), opioids were taken by nearly half of participants (*z* = 49%, (21%, 79%)) and NSAIDs were taken by approximately two thirds of patients (*z* = 72%, (39%, 91%)). Estimated marginal mean number of taken anticonvulsants was *M* = 5.35, (2.6, 11.19), for NSAIDs it was *M* = 9.41, (6.37, 14.44), and for opioids *M* = 5.46, (3.40, 8.45).

### 3.4. Oswestry Scores

Finally, we tested for significant changes in Oswestry scores after participants completed the study. Although the median Oswestry score was noticeably higher after the trial, Me (IQR) = 30.5 (10), as compared to before the trial, Me (IQR) = 19 (15), this effect did not reach significance, Z = −1.96, *p* = 0.056. This may be because complete data were available only for eight participants.

### 3.5. Complications

In one patient (female, age 74), a delayed allergic reaction with a negative wound culture at the site of the implanted stimulator was observed after 13 months. The unavoidable removal of IPG occurred in 4.7% (*n* = 3) of patients: (1) female, age 74 after 17 months; (2) male, age 72 after 10 months; and (3) male, age 65 after 12 months. Electrodes and IPGs were removed due to unsatisfactory pain relief in 13% of patients. In 1 case (male, 35), the IPG had to be replaced due to depleted battery after 10 months, and in another case (female, 43), the electrodes had to be replaced after 15 months due to electrode dysfunction (9.5%).

## 4. Discussion

One of the main advantages of this study was the double-blind setting in sub-perception modes and the placebo control. These settings were made possible by paraesthesia-free stimulation that was applied in this trial. No sensory perceptions were observed during the three allocated treatment arms (i.e., clustered tonic stimulation, 1 kHz stimulation, and sham stimulation). Furthermore, the crossover design allowed for each subject to be exposed to each modality, which allowed for individual subjects to serve as their own control. A benefit of the crossover design is an increase in statistical power even in this relatively limited sample size. Moreover, the physician examining each patient was not aware of the type of stimulation used, which mitigated any performance bias. Patients only felt paraesthesia during the tonic LF stimulation condition. Therefore, the present trial can be considered to be semi-blinded. In tonic mode patients were always aware of active stimulation and knew when it was switched off. Unfortunately, it has been demonstrated that the conventional mode of stimulation predominantly covered limb areas with insufficient pain relief in axial regions [3].

The primary goal of the study was to establish the type of stimulation was the most effective for relieving pain, and to establish which type was the most preferred in long-term stimulation. Patients did not report annoying paraesthesia during sub-perception modes. Patients were not aware of the type of stimulation that had been programmed and were not aware of whether the stimulation was on or off because they were informed that they could not perceive any sensations. There were no washout periods between subsequent treatment arms. Thus, each treatment arm ended the same day that the next arm was initiated.

The design of this cross-over study could be burdened by carry-over effects, but the analysis revealed that average pain was relatively stable through the entire study periods. A previous trial used a similar double-blind placebo-controlled design and examined preferred frequencies and waveforms [15]. That study used a 2-day washout period and did not identify lingering pain reductions caused by the prior intervention [15].

In the present study, average values of pain intensity after each type of stimulation were not shown to be superior to sub-perception stimulation. The observed reduction in pain was modest across all modalities. During the sham arm, devices showed that the IPG was on. It is possible that a small leaking current with maximal amplitude of 0.1 amp could have some therapeutic effect; however, the magnitude of this effect is likely negligible. The outcome can be attributed to the awareness that IPG was active to a greater extent than the real analgesic effect coming from stimulation with maximal amplitude of 0.1 Amp, f = 2 Hz, PW =10 μs. Nevertheless, we observed a strong placebo effect in this study. To exclude the potential influence of carryover effects from the previous treatment arm, average daily trends were assessed. We observed no significant difference in these daily trends. Results of the present study demonstrated a lack of persistent effects of earlier interventions. Probably there was a strong connection between implantation of SCS and improvement in psychological perception of pain presented in high placebo effect.

The reduction of pain after the application of all modes was statistically significant in relation to baseline values. Differences between various waveforms did not reach significance. LF tonic SCS had, however, the most robust analgesic effect of 50%. At the end of the randomised phase, the majority of subjects (55%) chose tonic stimulation as their preferred mode for SCS. Further, we found that LF tonic SCS was associated with the highest level of relative pain relief. Comparable results were reported in another study in which over 70% patients reported relief of low-back and/or limbs pain following tonic stimulation with 3D neural targeting [28]. In a randomised controlled double-blind cross-over study conducted by Wolter at al., supra-threshold tonic stimulation was associated with a higher degree of pain relief in comparison to sub-threshold tonic stimulation and a control arm without stimulation [16]. In that study, mean NRS score during a 1-week period without stimulation was *M* ± SD = 6.44 ± 2.04. In contrast VAS scores recorded in the present study during the 2-week trial period without stimulation were 5.43 ± 2.01. In the study by Wolter et al., there was no washout period [16].

In our study, we applied clustered tonic stimulation as a “burst stimulation” with 450–550 Hz in a cluster activated with f = 40–60 Hz. The main difference between BurstDR SCS invented by de Ridder and clustered tonic stimulation is that the former was designed to mimic burst neuronal firing. BurstDR also has the ability to activate medial pathways whereas clustered tonic stimulation likely does not [29]. It is possible that clustered tonic stimulation provides a weaker modulatory effect on afferent nerves, which may explain the lack of superiority of this technique over other stimulation techniques in the present study.

In contrast to prior studies, we did not find clustered tonic stimulation (i.e., burst) to be superior to other forms of stimulation. A prior randomised, placebo-controlled study demonstrated higher effectiveness of burst stimulation in comparison with 500 Hz tonic stimulation at subsensory amplitude and placebo stimulation [14]. Another randomised placebo controlled trial compared burst, tonic and placebo stimulation and found that burst stimulation was associated with the greatest reductions in back and limb pain [30]. Previous studies examining the effects of BurstDR stimulation demonstrated higher efficacy of BurstDR not only for reducing leg pain but also for reducing axial pain. The SUNBURST study provided evidence for higher efficacy of burst therapy in relation to tonic therapy [11]. A clinical review demonstrated a general superiority in efficacy of burst stimulation over tonic stimulation for reducing not only the somatic component of pain but also for addressing the emotional and psychological elements of pain [31]. A pooled analysis benefit of burst over tonic SCS was presented in that article. Further, a majority of subjects preferred burst stimulation over other types of stimulation due to resolution of emotional and cognitive aspects of pain [32].

Poor outcomes following clustered tonic and 1 kHz stimulation can be caused by insufficient coverage of the pain area and suboptimal electrode lead placement. In our sample, most patients had percutaneous lead placement with intraoperative trial stimulation to verify the coverage of pain areas. Therefore, we assume that electrodes were placed in an optimal location. In general, we did not observe migration of electrodes nor loss of paraesthesia in the area of pain. However, there was one case where electrode leads required replacement due to dysfunction. Furthermore, Van Buyten et al. observed that generators were explanted in 14% of patients with HF stimulation due to inadequate pain relief [33]. We did not observe higher efficiency of 1 kHz stimulation as compared to other forms of stimulation. This observation is in agreement with conclusions drawn from a prior review of HF stimulation clinical trials, which showed a lack of high quality evidence on the superiority of HF SCS [34]. Perruchoud et al. observed equal analgesic effect of 5 kHz stimulation and sham in the first double-blind crossover randomized study on SCS. The authors applied paresthesia-free HF stimulation and received similar, stable VAS scores through 2-week periods of sham and HF SCS, but better response was related to a “period effect” and order in the sequence of treatment modalities [35]. In the present study, the highest frequency that was applied was 1 kHz. For patients with percutaneous leads, the optimal SCS was assigned during the trial period using tonic stimulation. Further, 1 kHz was activated based on the active contact alignment. Of note, seven participants in the present study (38%) preferred this mode of stimulation. Several studies have shown that there is no need to increase the frequency to improve the therapeutic effect [13,18]. Similar outcomes were achieved after 1, 4, 7, and 10 kHz stimulation in the PROCO RC TRIAL, which demonstrated that these frequencies are equivalent in efficacy [13]. The WHISPER RCT provided clinical evidence that sub-perception stimulation with 1.2 kHz is non-inferior to supra-perception stimulation and can be effectively used to treat chronic intractable pain [17]. On the other hand, in the prospective, randomized, sham-control, double-blind, crossover trial of subthreshold SCS published by Al-Kaisy et al., the authors demonstrated that sham stimulation produced similar analgesic effect as 1.2 kHz and 3 kHz SCS, which is concordant with our results. The only one significantly effective in terms of pain relief was 5882Hz stimulation [36]. Interestingly, primary reports have demonstrated that response is optimal after 10 kHz stimulation. For example, the SENZA-RCT compared tonic and 10 kHz SCS and found that 80% of patients treated with HF stimulation reported better pain relief for back and/or leg pain, whereas 50% of patients reported pain relief following tonic SCS [12,37]. In contrast, in an observational, small cohort study designed to compare long-term effects (e.g., 20 months) of burst and 10 kHz modulation in FBSS patients, burst modality was found to be superior and more stable over time as compared to 10 kHz modulation [38]. In our study, however, we did not observe superiority of sub-perception stimulation over the supra-perception modality. This observation is consistent with conclusions drawn in an analogous study on various paradigms of SCS in patients with CRPS [15].

The true efficacy of SCS can be described by the number of therapy-related explants due to inadequate pain relief. In a multicentre study, the explant rate after SCS was 7.9% per year. Over half of these explants were due to failed analgesic effects, with 14.2% of those related to HF (10 kHz) rechargeable SCS [33]. Unfortunately, the incidence of explants was also high in the present study; indeed, for 3 of the 23 cases (13%), the stimulating system had to be removed following patients’ report of unsatisfactory improvement. This explant rate is extremely high and reflects a failure of treatment in the short-term. Data from a recent study show that rate of device removal in long term follow-up can be as high as 30% [39]. In one case, an inflammatory reaction occurred at the site of the IPG, which was caused by hypersensitivity and resulted in skin erosion over the device. Similar cases have been described by Chaudhry et al. [40].

### Limitations

The period of assessment stimulation lasted 2 weeks and may not be not long enough to assess longevity and effectiveness of each modality. Longer assessment periods may result in different scores. For example, in the SUNBURST study, the treatment arms lasted 12 weeks [11]; in the WHISPER trial, 3 months [17]; and in PROCO RCT [13], 4 weeks. In other trials, the crossover period took place after 2 weeks [14,15,35]. The present study employed a cross-over design with a placebo comparator; however, this type of study can be biased by participant expectations. Patients who reported improvement on LF tonic stimulation during the trial period when they felt paraesthesia may anticipate that these sensations would guarantee a successful treatment in the tonic phase. Recall bias may also be a potential factor wherein patients did not remember how strong the pain was preoperatively. In contrast, patients could remember how poor the effect was in the sham stimulation period as compared to the active stimulation period. Additional important limitations of this study include the relatively small number of subjects and high percentage of dropouts. However, the cross-over design may help to mitigate these limitations. Attrition bias may also occur in our study due to lack of satisfactory pain relief during treatment phases.

## 5. Conclusions

The present study demonstrated that the application of SCS had an impact on pain relief. The pain-relieving effects reached significance and were comparable across all modes of stimulation including sham. Sub-perception stimulation was not superior to supra-perception stimulation. In this study, SCS was characterised by a large placebo effect. The most preferred type of stimulation was the LF tonic modality, given that perceptions may reflect active stimulation. No evidence of carryover effect was observed between subsequent treatments with different frequencies. Contemporary neuromodulation procedures should be tailored to the individual preferences of patients to achieve satisfactory effects. Patients are encouraged to actively participate in selection of the most effective mode of stimulation.

## Figures and Tables

**Figure 1 jcm-09-02810-f001:**
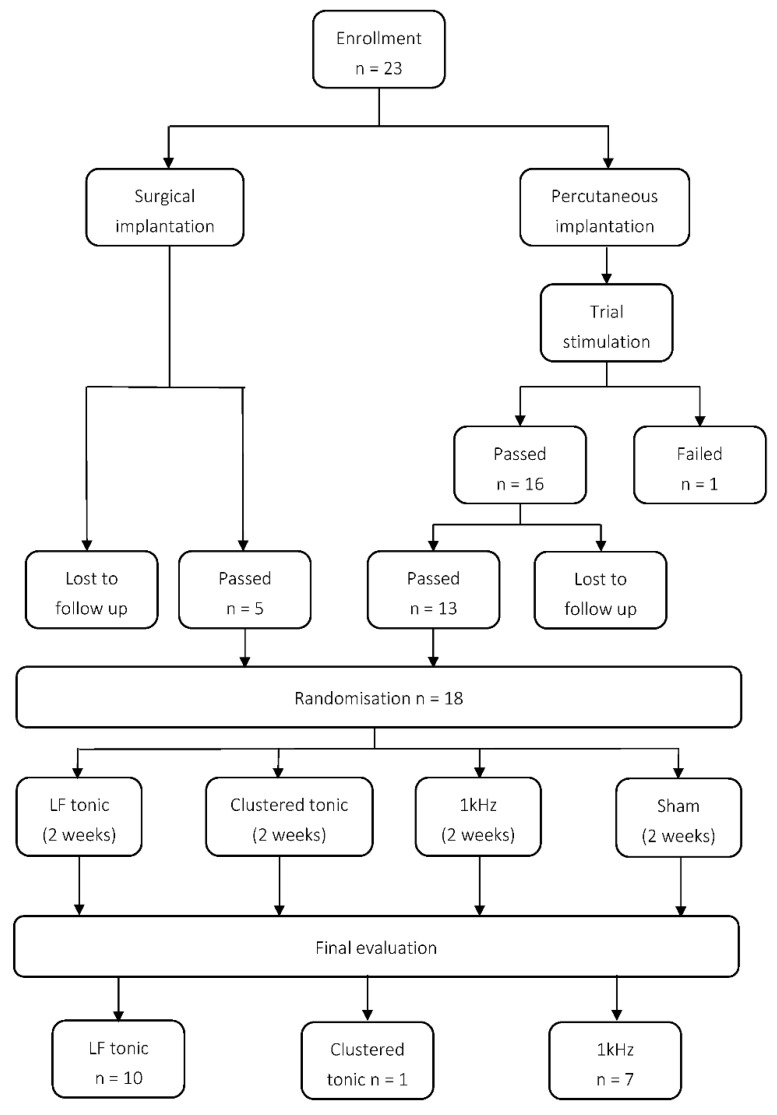
Consort flowchart.

**Figure 2 jcm-09-02810-f002:**
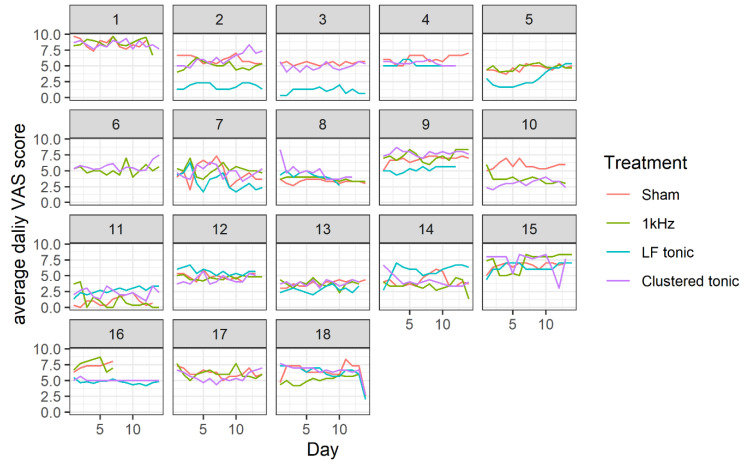
Average daily visual analogue scale (VAS) scores, separately for each treatment and subject, as a function of treatment day.

**Figure 3 jcm-09-02810-f003:**
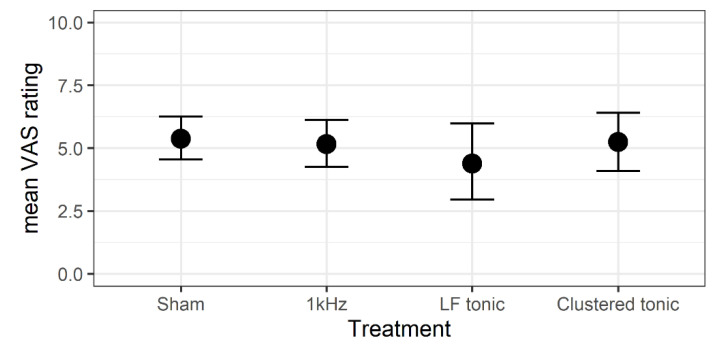
Predicted marginal means of mean VAS scores.

**Figure 4 jcm-09-02810-f004:**
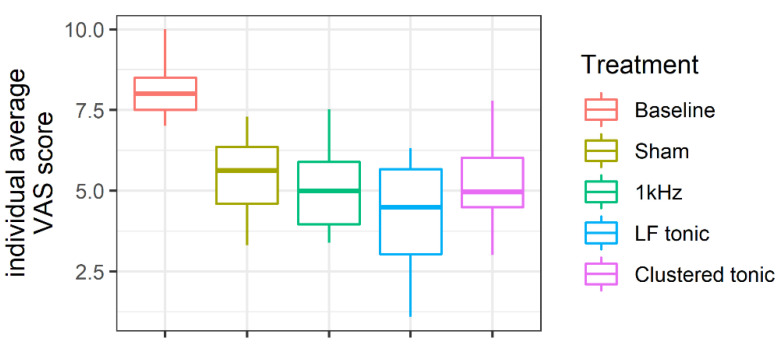
Distributions of individual average VAS scores for each treatment type and baseline.

**Figure 5 jcm-09-02810-f005:**
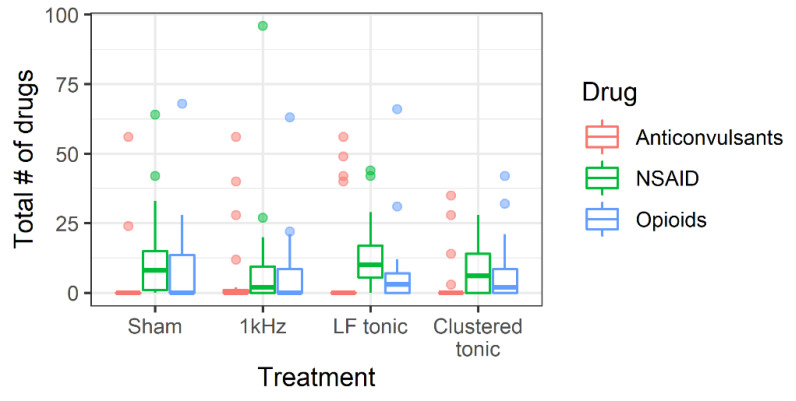
Distributions of total number of medications taken separated by medication and treatment types.

**Table 1 jcm-09-02810-t001:** Characteristics of patients. FBSS, Failed-Back Surgery Syndrome; CRPS, Complex Regional Pain Syndrome. Type of electrode: P, percutaneous; S, surgical; 1 × 8, eight contacts electrode; 2 × 8, two eight contacts electrodes; 16, 16 contacts electrode. PN, Precision Novi Boston Scientific non-rechargeable internal pulse generator; M, Montage Boston Scientific rechargeable internal pulse generator.

nr	Age	Sex	Diagnosis	Duration of Pain Years	Trial	Type of Electrode	Complication	VAS Baseline	Preferred Mode
*1*	72	m	FBSS	5	yes	P2x1x8 PN	no follow-up/after 10 months removal	9	-
*2*	48	f	FBSS	15	yes	P1x8 PN	no	8	LF
*3*	43	f	FBSS	10	yes	P1x16/P1x8 PN	replacement after 15 months/dysfunction	8	1kHz
*4*	50	m	CRPS	30	no	S2x8 PN	no	9	1kHz
*5*	52	m	FBSS	3	yes	P1x8 PN	no	9	LF
	58	m	FBSS	4	no	S2X8 PN	no	9	LF
*7*	35	m	FBSS	2	yes	P1x8 PN/M	replacement after 10 months/battery depletion	7	LF
*8*	59	f	FBSS	8	yes	P1x8 PN	no	8	LF
*9*	46	f	FBSS	4	yes	P1x8 PN	no	7	LF
*10*	62	m	CRPS	1	yes	P1x8 PN	no	10	LF
*11*	53	f	FBSS	11	no	S2x8 M	no	8	1kHz
*12*	60	m	FBSS	3	yes	P1x8 PN	no follow-up		
*13*	53	f	FBSS	9	yes	P2x1x8 PN	no	8	LF
*14*	65	m	FBSS	5	yes	P2x1x8 PN	removal after 12 months/no pain relief	7	1kHz
*15*	62	f	CRPS	10	no	S2x8 PN	no	9	1kHz
*16*	62	f	FBSS	9	no	S2x8 PN	no	8	Clustered
*17*	74	f	FBSS	15	yes	P2X1X8 PN	removal after 17 months/no pain relief	7	1kHz
*18*	44	m	FBSS	3	yes	P1x8 PN	no	10	LF
*19*	74	f	FBSS	4	yes	P2x1x8 PN	removal after 13 months/allergic reaction	10	1kHz
*20*	57	m	FBSS	6	yes	P2x1x8 PN	no	8	LF
*21*	62	m	CRPS	2	yes	P1x8	failed trial		
*22*	65	m	CRPS	10	no	S2x8 PN	no follow-up		
*23*	64	f	FBSS	8	yes	P2x1x8 PN	no follow-up		

**Table 2 jcm-09-02810-t002:** Results of one sample t-tests for ΔVAS scores, separately for each treatment type.

	VAS		ΔVAS		95% CI		*t*	*df*	*p*	Cohen’s *d*	% Pain Reduction
	*M*	*SD*	*M*	*SD*	LI	UI					
Sham	5.42	1.22	2.73	1.70	1.74	3.71	5.99	13	<0.001	1.66	0.34
1kHZ	5.17	1.40	3.04	1.47	2.20	3.89	7.76	13	<0.001	2.15	0.37
LF tonic	4.18	1.76	4.07	2.11	2.73	5.41	6.68	11	<0.001	2.02	0.50
Clustered tonic	5.27	1.33	2.80	1.63	1.86	3.74	6.42	13	<0.001	1.78	0.34
Baseline	8.13	0.99									

Note: *p* values are after Bonferroni correction.

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
