# Peer review of "Sub-Perception and Supra-Perception Spinal Cord Stimulation in Chronic Pain Syndrome: A Randomized, Semi-Double-Blind, Crossover, Placebo-Controlled Trial"

_jcm, 2020, doi:10.3390/jcm9092810_

Round 1
Reviewer 1 Report
The authors addressed my concerns in the revised version of the manuscript.
Author Response
We thank for important comments and suggestions .
Thank you very much for your acceptance of the revised version.
With best regards
The authors
Reviewer 2 Report
The present study by Sokal et al. sought to evaluate the effectiveness of low and high frequency SCS among patients with chronic pain using a randomized, cross-over study design. It is reported that the pain-relieving effects of SCS were comparable across all modes of stimulation including sham, and that low frequency was the most preferred by patients. This study addresses an important clinical question and would contribute the body of literature regarding neuromodulation for chronic pain. However, there are several concerns that should be addressed.
- During randomization, all patients are aware that the stimulator is either low frequency or not low frequency because of paresthesia. Therefore, this is not a blinded or semi-blinded study, and does not include a true placebo condition. Please consider revising because this is very important for interpretation of the results.
- All patients went through trial stimulation, but the CONSORT flowchart shows a portion not going through the trial stimulation. Also, there were 18 patients randomized, but Figure 2 show 19 patients. Please clarify and revise if needed.
- Was the coverage of pain area checked in the seated position at the beginning of each arm (LF, clustered tonic, 1kHz, and sham)? This should be clarified in the manuscript.
- Patients received tonic LF stimulation twice, 2 weeks of trial stimulation and 2 weeks in the randomization. Did pain scores differ between the trial stimulation and the randomization using tonic LF? This may reveal important information about reproducibility and may also lend further support to the finding that tonic LF produced the largest reduction in pain ratings.
- In addition to SCS parameters, pain relief with SCS may be predicted by patient characteristics. For example, the experience of pain has shown to be different between men and women. Also, there is a large overlap between cardiovascular and pain regulatory systems. Were blood pressure measurements taken and did these correlated with pain in this study? Any information that can improve the prediction of SCS success is needed. Therefore, any patient characteristics that predict or correlate with reduced pain ratings should be reported.
- The literature review in the discussion section is difficult to follow at times. For example, studies showing greater efficacy using high frequency SCS are scattered and discussed individually. Organizing the literature in sections based on outcomes and providing summaries would be helpful to the reader.
- Line 714: “The period of assessment stimulation lasted 2 weeks and may not be long enough to assess longevity and effectiveness of each modality. Longer assessment periods may result in different scores.” Why would 2 weeks not be long enough (this was the chosen study protocol) and why would scores be different with longer assessments? This should be discussed in the context of published studies.
- Line 716: Patients who were in the first phase in the placebo group may have lower expectations during the stimulation phase, whereas patients who had active treatment during the first phase may anticipate continued improvement in the placebo phase.” Although this may be true, the impact on the results should be minimized because the study was randomized. This statement should be revised or removed.
- There is discrepancy in the interpretation of the sham condition. Line 103 in the abstract states that there was “no evidence of carryover effect between subsequent treatments,” but Line 594 states that the authors “suspected that the reduction of pain during sham stimulation might arise from a carryover effect of active stimulation from the previous treatment arm.” This needs to be revised.
Author Response
We thank for important comments and suggestions, which helped us improve and revise our manuscript.
- During randomization, all patients are aware that the stimulator is either low frequency or not low frequency because of paresthesia. Therefore, this is not a blinded or semi-blinded study, and does not include a true placebo condition. Please consider revising because this is very important for interpretation of the results
In my opinion patients were aware that the stimulation was on only in period with low frequency tonic stimulation, but in periods of sham stimulation (sham), clustered tonic stimulation, which corresponds with burst stimulation with sub-perception level of amplitude, and when stimulation was in high-frequency mode (1,2kHz) with sub-perception level - they did not feel the paresthesia. Therefore, in these 3 arms: sham,1kHz and clustered tonic SCS participants were blinded and did not know, what stimulation was on and in what sequence. Patients were aware of stimulation in the trial external stimulation conducted at least two weeks prior to randomisation to one of four treatment arms. No examining physician nor patients knew in what arm they were during the study, apart from period of LF when they usually felt paresthesia. That’s why the trial was semi-blinded. It is clarified in the discussion:
One of the main advantages of this study was the double-blind setting in sub-perception modes and the placebo control. These settings were made possible by paraesthesia-free stimulation that was applied in this trial. No sensory perceptions were observed during the three allocated treatment arms (i.e., clustered tonic stimulation, 1 kHz stimulation, and sham stimulation). Furthermore, the crossover design allowed for each subject to be exposed to each modality, which allowed for individual subjects to serve as their own control. A benefit of the crossover design is an increase in statistical power even in this relatively limited sample size. Moreover, the physician examining each patient was not aware of the type of stimulation used, which mitigated any performance bias. Patients only felt paraesthesia during the tonic LF stimulation condition. Therefore, the present trial can be considered to be semi-blinded.
- All patients went through trial stimulation, but the CONSORT flowchart shows a portion not going through the trial stimulation. Also, there were 18 patients randomized, but Figure 2 show 19 patients. Please clarify and revise if needed.
I thank the reviewer for this important remark. The figure nr 2 was revised and updated. The number of patients examined after randomization was 18. It corresponds with the CONSORT flowchart. Exclusion of one participant which did not belong to the group of 18 did not significantly change the results of the study.
Sentence with the results in the abstract, tables and figures were updated.
- Was the coverage of pain area checked in the seated position at the beginning of each arm (LF, clustered tonic, 1kHz, and sham)? This should be clarified in the manuscript.
I clarified this issue in the section:
2.3. programming and randomization
…
Participants were programmed in a sitting position without checking the coverage area, given that the coverage check was already conducted in an external trial stimulation 2 weeks prior to randomisation.
- Patients received tonic LF stimulation twice, 2 weeks of trial stimulation and 2 weeks in the randomization. Did pain scores differ between the trial stimulation and the randomization using tonic LF? This may reveal important information about reproducibility and may also lend further support to the finding that tonic LF produced the largest reduction in pain ratings.
We did not include the results of the trial LF tonic stimulation before randomisation into the study. The objective of this trial stimulation with external stimulator was to assess the coverage of pain and check if the analgesic effect was satisfactory - over 50% reduction of pain. Subsequently patients underwent permanent IPG implantation
We wanted to compare four modes of SCS (LF, 1kHz, clustered, sham) in a trial which actually started after randomisation in patients with permanent IPGs
- In addition to SCS parameters, pain relief with SCS may be predicted by patient characteristics. For example, the experience of pain has shown to be different between men and women. Also, there is a large overlap between cardiovascular and pain regulatory systems. Were blood pressure measurements taken and did these correlated with pain in this study? Any information that can improve the prediction of SCS success is needed. Therefore, any patient characteristics that predict or correlate with reduced pain ratings should be reported.
Unfortunately, we did not measure blood pressure during the trial, only before the implantation of neurostimulator, we did not examine correlation of cardiovascular parameters and effects of SCS. We will take into account this suggestion in future studies. No correlation between sex and efficacy of SCS was found in our study basing on the preliminary analysis and Mann-Whitney test :f/m baseline p=0,817, LF tonic p=0,534, clustered p=0,306, 1kHz p=0,894, sham p=0,450
- The literature review in the discussion section is difficult to follow at times. For example, studies showing greater efficacy using high frequency SCS are scattered and discussed individually. Organizing the literature in sections based on outcomes and providing summaries would be helpful to the reader.
We checked and looked over the text according reviewer's suggestion
We tried to summarize the positions of literature in the best way we could in a very concise style especially concerning important cited studies on high frequency SCS. Further extension of these descriptions would however enlarge the discussion to an unacceptable size. We did some corrections visualized in revised version of the manuscript
We did not observe higher efficiency of 1kHz stimulation as compared to other forms of stimulation. This observation is in agreement with conclusions drawn from a prior review of HF stimulation clinical trials, which showed a lack of high quality evidence on the superiority of HF SCS [34]. Perruchoud et al. observed equal analgesic effect of 5kHz stimulation and sham in the first double-blind crossover randomized study on SCS. The authors applied paresthesia-free HF stimulation and received similar, stable VAS scores through 2 weeks periods of sham and HF SCS but better response was related to a “period effect” and order in the sequence of treatment modalities [35]. In present study, the highest frequency that was applied was 1 kHz. For patients with percutaneous leads, the optimal SCS was assigned during the trial period using tonic stimulation. Further, 1kHz was activated based on the active contact alignment. Of note, seven participants in the present study (38%) preferred this mode of stimulation. Several studies have shown that there is no need to increase the frequency to improve the therapeutic effect [13,18]. Similar outcomes were achieved after 1, 4, 7, and 10 kHz stimulation in the PROCO RC TRIAL, which demonstrated that these frequencies are equivalent in efficacy [13]. The WHISPER RCT provided clinical evidence that sub-perception stimulation with 1.2 kHz is non inferior to supra-perception stimulation and can be effectively used to treat chronic intractable pain [17]. On the other hand, in the prospective, randomized, sham-control, double-blind, crossover trial of subthreshold SCS published by Al-Kaisy et al., the authors demonstrated that sham stimulation produced similar analgesic effect as 1.2kHz and 3kHz SCS what is concordant with our results. The only significantly effective in terms of pain relief was 5882Hz stimulation [36]. Interestingly, primary reports have demonstrated that response is optimal after 10 kHz stimulation. For example, the SENZA-RCT compared tonic and 10 kHz SCS and found that 80% of patients treated with HF stimulation reported better pain relief for back and/or leg pain, whereas 50% of patients reported pain relief following tonic SCS [12] [37]. In contrast, in an observational, small cohort study designed to compare long-term effects (e.g. 20 months) of burst and 10 kHz modulation in FBSS patients, burst modality was found to be superior and more stable over time as compared to 10 kHz modulation [38]. In our study, however, we did not observe superiority of sub-perception stimulation over the supra-perception modality. This observation is consistent with conclusions drawn in an analogous study on various paradigms of SCS in patients with CRPS [15].
- Line 714: “The period of assessment stimulation lasted 2 weeks and may not be long enough to assess longevity and effectiveness of each modality. Longer assessment periods may result in different scores.” Why would 2 weeks not be long enough (this was the chosen study protocol) and why would scores be different with longer assessments? This should be discussed in the context of published studies.
In limitations, which were listed we decided to point out that period of stimulation lasting 2 weeks may be to short to assess objectively effectiveness of each modality basing on results of other studies which showed that true effect of high frequency stimulation or burst stimulation might be seen after longer period.
For example, in the SUNBURST study, the treatment arms lasted 12 weeks [11]; in the WHISPER trial, 3 months [17]; and in PROCO RCT [13], 4 weeks. In other trials, the crossover period took place after 2 weeks [14] [15] [35].
- Line 716: Patients who were in the first phase in the placebo group may have lower expectations during the stimulation phase, whereas patients who had active treatment during the first phase may anticipate continued improvement in the placebo phase.” Although this may be true, the impact on the results should be minimized because the study was randomized. This statement should be revised or removed.
I thank for this comment I agree that this statement is not necessary, describes theoretical situation and could be removed.
- There is discrepancy in the interpretation of the sham condition. Line 103 in the abstract states that there was “no evidence of carryover effect between subsequent treatments,” but Line 594 states that the authors “suspected that the reduction of pain during sham stimulation might arise from a carryover effect of active stimulation from the previous treatment arm.” This needs to be revised.
It is true that we suspected analgesic effect carried over from previous treatment arm during sham but analysis of the results of VAS score in subsequent days of 2 week sham stimulation revealed that this difference was not significant
We decided to change this sentence according to reviewer’s suggestion:
To exclude the potential influence of carryover effects from the previous treatment arm, average daily trends were assessed. We observed no significant difference in these daily trends. Results of the present study demonstrated a lack of persistent effects of earlier interventions.
The paper has been revised by an English Editor.
Thank You very much for valuable remarks. They were very professional and meritorical.
With best regards
The authors
This manuscript is a resubmission of an earlier submission. The following is a list of the peer review reports and author responses from that submission.
Round 1
Reviewer 1 Report
Given that patients were only in each treatment arm for 2 weeks it appears to be difficult to assess the full efficacy of each treatment model as reprogramming for some modalities takes a significant amount of time. My guess would be that LF would be of preference as parasthesia mapping would ensure that there was proper coverage immediately along with detecting sensation to ensure it was on. Additionally, as mentioned previously there is no washout period which makes it unclear if there was residual neuromodulatory effects.
I think it would also be beneficial to see more detailed explanation regarding pain medications in addition to the graft. For example, where opioids given for post operative pain control during trials or implants? Did post operative pain management differ between percutaneous and paddle leads?
I did find it interesting that your study demonstrated minimal benefit for burst stimulation which was unexpected.
Reviewer 2 Report
In this interesting manuscript, Sokal et al. report a randomized crossover trial comparing four SCS waveforms in 18 patients with failed back surgery syndrome and complex regional pain syndrome. They compared conventional paresthesia-based tonic stimulation, burst, 1 kHz, and sham for two weeks. They found that all types of stimulation improve pain better than before stimulation and that there was no significant difference between groups. Most patients preferred the conventional tonic paradigm, where they felt the stimulation.
Major issues:
I can't entirely agree with the conclusion that "SCS was effective in pain relief" (Abstract, line 42). SCS was no better than sham stimulation; therefore, the author's conclusion is not aligned with their findings. It is unclear how SCS efficacy was inferred as there were no differences between groups for pain relief, medications used, and Oswestry scores. The manuscript should be re-written to reflect the results of the study better. Other prior publications, such as Al-Kaisy, Neuromodulation, 2018, found that 1.2 kHz is no different from sham stimulation, in a very similar study (also see Perruchoud 2013). That reference should be cited.
There is some discussion about the strong placebo effect found in this study (P11, L292). Very few trials have compared active with sham SCS, despite being technically feasible. This is an essential finding of the study, and it should be further addressed.
Minor:
1) The description of the burst stimulation algorithm is too vague:
"intermittent packets of burst stimuli delivered using the neural targeting algorithm, which consisted of several pulses per packet with PW 250-500 μs repeated with f = 40 Hz". What is the neural targeting algorithm? Did all patients get the same number of pulses per burst? How were the contacts used chosen? There is information in the discussion (p 12, L 325) that could be added to the methods.
2) Table 1 needs to have the abbreviations defined.
3) The authors should be cautious when describing the type of burst used in this study with BurstDR – a non-charge balanced waveform. This nomenclature is a source of significant confusion in the field, and it is this reviewer's opinion that the two waveforms are not equivalent in efficacy and mechanisms of action. I would suggest removing the word 'burst' entirely from the manuscript, and referring to it as clustered tonic stimulation, as done on P12, L327.
4) Similarly, the HF stimulation used in this study is easily confused with HF10. I suggest that HF is replaced with 1 kHz stimulation to clarify the issue.